# The Influence of Magnetic Composite Capsule Structure and Size on Their Trapping Efficiency in the Flow

**DOI:** 10.3390/molecules27186073

**Published:** 2022-09-17

**Authors:** Roman Verkhovskii, Alexey Ermakov, Oleg Grishin, Mikhail A. Makarkin, Ilya Kozhevnikov, Mikhail Makhortov, Anastasiia Kozlova, Samia Salem, Valery Tuchin, Daniil Bratashov

**Affiliations:** 1Science Medical Center, Saratov State University, 83 Astrakhanskaya Str., 410012 Saratov, Russia; oualeksej@yandex.ru (A.E.); discovery-line@mail.ru (O.G.); makmih50@gmail.com (M.A.M.); kozhevnikov_io@mail.ru (I.K.); mmahortv@mail.ru (M.M.); anastasiia.kozlova245@gmail.com (A.K.); samiafarouk66@yahoo.com (S.S.); tuchinvv@mail.ru (V.T.); dn2010@gmail.com (D.B.); 2Institute of Molecular Theranostics, I. M. Sechenov First Moscow State Medical University, 8 Trubetskaya Str., 119991 Moscow, Russia; 3Department of Physics, Faculty of Science, Benha University, Benha 13511, Egypt; 4Laboratory of Laser Molecular Imaging and Machine Learning, Tomsk State University, 36 Lenin’s Ave., 634050 Tomsk, Russia; 5Institute of Precision Mechanics and Control, FRC “Saratov Scientific Centre of the Russian Academy of Sciences”, 24 Rabochaya Str., 410028 Saratov, Russia; 6Bach Institute of Biochemistry, FRC “Fundamentals of Biotechnology of the Russian Academy of Sciences”, 119071 Moscow, Russia

**Keywords:** targeted drug delivery, polyelectrolyte microcapsules, magnetic drug delivery, magnetic separation, flow cytometry, internalization of microcapsules, magnetic cell sorting

## Abstract

A promising approach to targeted drug delivery is the remote control of magnetically sensitive objects using an external magnetic field source. This method can assist in the accumulation of magnetic carriers in the affected area for local drug delivery, thus providing magnetic nanoparticles for MRI contrast and magnetic hyperthermia, as well as the magnetic separation of objects of interest from the bloodstream and liquid biopsy samples. The possibility of magnetic objects’ capture in the flow is determined by the ratio of the magnetic field strength and the force of viscous resistance. Thus, the capturing ability is limited by the objects’ magnetic properties, size, and flow rate. Despite the importance of a thorough investigation of this process to prove the concept of magnetically controlled drug delivery, it has not been sufficiently investigated. Here, we studied the efficiency of polyelectrolyte capsules’ capture by the external magnetic field source depending on their size, the magnetic nanoparticle payload, and the suspension’s flow rate. Additionally, we estimated the possibility of magnetically trapping cells containing magnetic capsules in flow and evaluated cells’ membrane integrity after that. These results are required to prove the possibility of the magnetically controlled delivery of the encapsulated medicine to the affected area with its subsequent retention, as well as the capability to capture magnetically labeled cells in flow.

## 1. Introduction

In a pathological state, the blood contains important prognostic markers ranging from circulating DNA to exosomes and circulating tumor cells (CTCs), the detection of which helps in predicting morbidity [1,2,3]. In this regard, a number of approaches have recently been developed to perform the detection and sorting of these objects [4]. The approach based on the selective binding of magnetic objects conjugated with antibodies to target prognostic markers seems to be one of the most promising since it provides a wide variety of configurations. A striking example of this approach is the CellSearch system based on anti-EpCAM coated magnetic particles, which bind with CTCs spread by the breast tumor and separate them from the whole blood sample [5]. However, all currently existing kits involve blood sampling which significantly reduces the sensitivity of such an approach since the average volume of blood samples is about 10 mL, which is more than 400 fold less than the total blood volume [6]. Moreover, prognostic markers are extremely rare objects that further obstruct their detection [4,7].

In recent years, the concept of detecting and isolating prognostic markers from the bloodstream by binding them with systemically administered magnetically sensitive objects has attracted the attention of researchers [8,9]. In general, this concept implies the injection of magnetically sensitive objects capable of selectively binding prognostic markers into the circulatory system, their long-term circulation, interaction with such markers, and following detection/magnetic separation from the blood. 

Also, magnetically sensitive objects have been considered a highly effective tool in solving such a challenging task as reducing side effects from drug applications drawing on the example of polyelectrolyte microcapsules. Over the past decades, a vast amount of studies have shown the high potential of polyelectrolyte magnetic microcapsules’ application as containers for remotely controlled drug delivery [8,10,11,12,13]. This concept is based on the idea of the systemic administration of magnetically sensitive drug-loaded micro-containers and their targeting of the affected area using an external magnetic field source [14].

Both of these concepts, namely the separation of prognostic markers and controlled drug delivery, involve the capture of magnetically sensitive objects from the blood flow and impose a number of requirements for them including high biocompatibility, long-term circulation in the bloodstream, a capability to avoid immune recognition with subsequent clearance and filtration in capillaries, and a high magnetic moment. In this regard, magnetic polyelectrolyte microcapsules synthesized from biocompatible polymers are a highly promising system [15,16]. Such microcapsules do not cause the cytotoxic and hemotoxic effect [11], and, unlike solid particles, hollow microcapsules have specific mechanical properties and can be easily deformed in the blood flow, which allows their systemic administration without obstructing blood supply in the crucial organs [17]. Also, polyelectrolyte microcapsules can be shielded from immune clearance by modifying their surface charge, which has been previously demonstrated by Merkle and co-authors in [18]. The high magnetic moment of such objects is provided through the highly efficient loading of the capsules with magnetite nanoparticles achieved using the freezing-induced loading (FIL) approach [19]. However, despite the above parameters being crucial for both the successful drug delivery and separation of prognostic markers, the microcapsule size itself can imply some additional limitations. A larger carrier size can lead to a faster microcapsule filtration from the bloodstream [11], decreasing the circulation time and altering the effectiveness of the drug delivery. Moreover, the encapsulation process is limited to mild conditions during the microcapsule synthesis, thus the active substances can’t be loaded into microcapsules with the use of organic solvents or mechanical stress [20]. 

At present, a number of articles considering the trapping of magnetically sensitive objects in flow have been published. The majority of these articles are dedicated to the study of magnetic nanoparticles’ behavior in the bloodstream under the influence of a permanent magnetic field and modeling of their magnetic trapping [21,22,23,24,25]. A considerably smaller number of studies are devoted to the behavior of submicron- and micron-sized objects. Parak and co-workers were one of the first to demonstrate the possibilities of the magnetic trapping of polymeric microcapsules functionalized with magnetic nanoparticles in the fluid flow [26]. Later, Voronin et al. demonstrated the magnetic trapping ability of multilayer composite magnetic microcapsules in blood flow in vitro and in vivo [14]. In one of the latest studies, Gorin and co-workers modeled the behavior of submicron magnetic capsules in blood flow depending on the capsule’s architecture. However, despite the sufficient number of such studies, there is a shortage of experimental data describing the magnetic capture of magnetically sensitive objects from the flow, and this process has not been adequately studied yet.

Previously, we investigated the dynamics of magnetic microcapsules aggregation in whole blood [27], evaluated their hemotoxicity, the duration of circulation in the vascular system, and biodistribution depending on the injected concentration [17] and capsule size [11], which are crucial factors for their safe application as an agent for systemic administration. We also estimated the efficiency of magnetic microcapsules’ trapping by the external magnetic field source depending on their size and structure in vitro [11] and demonstrated the ability to separate systemically administered microcapsules from the bloodstream of laboratory animals in vivo [8]. These results indicate the prospectivity of using microcapsules as a tool for remotely controlled drug delivery and the isolation of prognostic markers. Here, we investigated the magnetic capture of polyelectrolyte microcapsules characterized by different magnetic moments from the flow in depth and evaluated the efficiency of their trapping depending on the flow velocity in vitro. Additionally, we verified the possibility of magnetically separating cells that internalized magnetic microcapsules from the flow in vitro and evaluated their viability after that.

## 2. Results

A magnetic field is one of the promising stimuli for remote noninvasive control over the drug delivery systems circulating in the bloodstream due to its deep penetration into soft tissues without significant attenuation [28]. However, magnetic force, as any stimuli, has safety limitations (≤3T static magnetic field for routine clinical application) [29,30] that impose some restrictions on the development of drug delivery systems including their sensitivity to the magnetic field. Moreover, the efficiency of magnetically capturing targeted objects from a flow largely depends on a range of properties including size, weight, and magnetic moment. However, these dependencies are difficult to extrapolate from the data obtained in static conditions. Hence, here we provide a detailed study of the magnetic capture efficiency of polyelectrolyte microcapsules from the liquid flow depending on the capsule size, magnetite nanoparticles (MNPs) payload, and flow velocity, which is crucial for the further introduction of the magnetic targeting approach to biomedical practice.

At first, we evaluated the magnetic properties of 1 ± 0.2, 2.7 ± 0.4 and 5.5 ± 0.8 μm (PSS/PAH)(PSS/BSA-FITC)/(PSS/PAH)_2_PSS capsules made with vaterite templates of the corresponding size loaded with MNPs using one, two, and three freezing/thawing cycles with the following template dissolution using the magnetic force microscopy (MFM) method. MFM images demonstrated an increasing magnetic moment between the composite polyelectrolyte microcapsule surface and cantilever following an increase in the number of freezing/thawing cycles applied to vaterite templates (Figure 1) that caused an expansion in the number of loaded MNPs. This is consistent with the data presented by Dmitry A. Gorin’s team [19,31], where they demonstrated an increase in the mass fraction of MNPs loaded into micron-sized vaterite particles following one, two, and three freezing/thawing cycles from 1.6% for one cycle to 4.6% and 6.4% for two and three cycles, respectively. Additionally, the individual capsule’s magnetic force increases following the enlargement of the capsule’s size, which was noted for two and three freezing/thawing cycles (Figure 1). The possible reason for this is that a larger vaterite template has a greater internal volume (total pore volume) and surface area resulting in a higher payload capacity per template, which facilitates the loading and sorbing of more MNPs [32,33].

Thus, capsules characterized by a high payload capacity and containing the maximal number of MNPs are more sensitive to the magnetic field and are capable of carrying plenty of drugs, which makes them the most suitable candidate for magnetically controlled drug delivery through systemic administration. However, the enlargement of such systems’ size leads to the shortening of their circulation period in the blood circulatory system [11]. Moreover, based on Stokes’ law (1) [34] the extension of the capsules’ size makes them more sensitive to the viscous drag force of the blood flow (Figure 2b), which also imposes some limitations on the design of drug delivery systems. Stokes’ law can be described by the following equation:
(1)
$$\vec{F_{D}}=-6\pi \eta R\vec{v}$$

where 
η
 is the fluid’s viscosity, *R* is the capsule’s radius, and 
$\vec{v}$
 is the capsule’s speed.

Another force determining the behavior of the magnetic polyelectrolyte capsule when passing through the vessel with an applied magnet is the magnetic force (Figure 2b), which can be described by the following equation [35]:
(2)
$$\vec{F_{M}}=\left(\vec{m}\;\cdot \;\nabla \right)\;\vec{B}$$

where 
$\vec{m}$
 is the magnetic moment, and 
$\vec{B}$
 is the magnetic field vector. The magnetic moment of a single magnetic capsule can be defined as follows:
(3)
$$\vec{m}=\rho V\left(\vec{M_{0}}\;\cdot \;\frac{\chi_{cap}}{\rho \mu_{0}}\;\vec{B}\right)$$

where, 
ρ,V
 are the density and volume of a capsule, 
$\vec{M_{0}}$
 is initial magnetization, 
$\chi_{cap}$
 is magnetic susceptibility of the capsule, and 
$\mu_{0}$
 is vacuum magnetic permeability. Therefore, the expanded form of the magnetic force’s equation is

(4)
$$\vec{F_{M}}=\rho V\left(\vec{M_{0}}\;\cdot \;\nabla \right)\;\vec{B}+\frac{V_{\chi_{cap}}}{\mu_{0}}\left(\vec{B}\;\cdot \;\nabla \right)\;\vec{B}$$


Based on this, the force acting on the capsule during its magnetic trapping in the flow 
$\left(\vec{F_{act}}\right)$
 can be presented as a sum of magnetic and viscous drag forces as follows:
(5)
$$\vec{F_{act}}=\vec{F_{D}}+\vec{F_{M}},$$


Successful capsules capture by the magnetic field occurs if the trajectory of the microcapsule under the influence of this combined force ends on the edge of the flow with zero velocity (Figure 2b). Otherwise, the capsule would slow down and change the trajectory of movement but would not stop in the area of interest.

To evaluate the influence of the viscous drag force of the liquid flow on the magnetic capture efficiency of differently sized capsules, we used the custom-built, SPIM-Fluid based flow cytometer (Figure 2a). Among all sizes of capsules loaded with MNPs by one freezing/thawing cycle, the largest one was found to be the most sensitive to the magnetic field and could be captured at flow velocity up to 100 mm/s. However, the number of captured capsules was insignificant and decreased following flow acceleration (Figure 3, first column-third line). The same tendency was observed for two freezing/thawing cycles (Figure 3, middle column-third line). However, at the maximal concentration of MNPs loaded into the capsules, 2.7 μm carriers demonstrated the best result (Figure 3, last column-second line). Hence, we found that the increase in the freezing/thawing cycle number and thus the amount of loaded MNPs leads to the enhancement of capsules’ magnetic moment for all sizes which corresponds with the MFM data. However, despite of the strongest magnetic properties of 5.5 μm capsules loaded through three freezing/thawing cycles, the efficiency of magnetic capture of 2.7 μm carriers was higher. This may be caused by the size and the weight gain of 5.5 μm capsules containing the largest number of MNPs in comparison with those weighing 2.7 μm. If we consider the forces acting on the capsule in terms of Newton’s second law, the resulting acting force can be presented as follows:
(6)
$$\vec{F_{act}}=m\frac{d\vec{v}}{dt}=-6\pi \eta R\vec{v}+\rho V\left(\vec{M_{0}}\;\cdot \;\nabla \right)\;\vec{B}+\frac{V_{\chi_{cap}}}{\mu_{0}}\left(\vec{B}\;\cdot \;\nabla \right)\;\vec{B},$$

where 
m
 is a capsule’s mass. Since the force acting on the capsule in the flow is in direct ratio to its mass (6), the existence of a mass cut-off value is expected, behind which the magnetic trapping of the capsule is impossible at the fixed strength of the magnetic field. Therefore, the magnetic capture of 5.5 μm capsules loaded through three freezing/thawing cycles is less efficient than that of 2.7 μm capsules since capsules’ weight and size gain results in an increase in the viscous drag force 
$F_{D}$
, and consequently, 5.5 μm capsules’ trapping requires the application of a higher magnetic force.

Another challenging issue of modern medicine is the separation of rare objects of interest, including diagnostic and prognostic disease markers from the whole blood [4]. In some cases, such objects are presented by pathological cells circulating in the vascular system [36,37] that can be magnetically trapped using selective binding with magnetically sensitive objects including nanoparticles [38], lipid nano-vehicles [39], and capsules [40]. The interaction between magnetic objects and target cells can occur in the following two ways: first, the object conjugated with antibodies can immunohistochemically (IHC) bind to the antigen on the cells’ surface [38,39,40], and second, the object can be internalized by a cell [11].

Nowadays, existing kits for the magnetic separation of cells implicate their segregation in static conditions via blood sampling and its following mixing with antibody-bonded magnetic labels. Since the blood sample volume that can be safely taken from a patient is around 10 mL [6], blood sampling dramatically reduces the sensitivity of such an approach. A much more effective way is the trapping of target cells from the bloodstream which avoids volume restriction [8,9]. However, the implementation of this approach is more complicated for a few reasons. First, it requires the injection of a tremendous number of magnetic labels into the bloodstream since the blood, beside target cells, contains many red blood cells (RBCs), white blood cells (WBCs), and platelets that make the interaction of magnetic labels with target cells less favorable. Second, the cell has a larger size and weight compared to the capsule, and the 
$\vec{F_{D}}$
 affecting the cell in dynamic conditions is much stronger. Thus, cell trapping requires the application of a stronger magnetic field or an increase in the number of magnetic labels linked with the cell. Moreover, IHC-bound labels on the cell surface probably play the role of a sail in the media flow that will amplify the effects of viscous drag force. Additionally, in a fluid stream, the essential role is probably played by the IHC binding force. Weak binding can result in the breaking of such bonds under media flow influence and consequently the removal of magnetic labels from the cells’ surface. To verify the ability to magnetically trap of cells from the flow, we simplified the model and excluded random variables. Thus, we considered the internalization as the result of the capsule-cell interaction to avoid the sail-effect and exclude the random possibility of IHC bond breaking.

The main part of blood cells, including RBCs and platelets, are not capable of internalizing micron-sized magnetic labels [41,42]. Thus, cells competing for the internalization of magnetic labels in the blood are represented by different subpopulations of WBCs normally contained in the blood and some pathological cells such as circulating tumor cells (CTCs). Here, we compared the efficiency of internalizing differently sized magnetic polyelectrolyte microcapsules by cells of metastatic cancer that can be presented in the blood as CTCs and by macrophage cells characterized by one of the highest levels of phagocytic activity. 4T1 cells were used as a model cancer cell line and Raw 264.7 as macrophages (Figure 4). The efficiency of microcapsules’ capture by the Raw 264.7 cell line was described in our previous article (Figure 4a) [11].

We have found that around 71% of Raw 264.7 cells during co-incubation with 1 μm capsules and 82% for 2.7 μm interacted with three or more capsules and internalized them during the first three hours (Figure 4a), while just around 7% of 4T1 cells during the co-incubation with 1 μm carriers and 16% for 2.7 μm interacted with three or more capsules, respectively (Figure 4b). Further observation showed no statistically significant increase in the number of microcapsules captured by Raw 264.7 cells after 6, 12 and 24 h of incubation independent of size. In contrast, for the 4T1 cell culture, we found a gradual increase in the number of cells capturing three or more capsules over time which rose to 39.5 ± 1.8% for 1 μm carriers and 31.9 ± 1.4% for 2.7 μm after 24 h of incubation, respectively. Therefore, we can conclude that macrophages capture and internalize microcapsules more actively in comparison with 4T1 cells when modeling CTCs. This may be caused by the higher phagocytic activity and motility of the macrophage cell line in comparison with the breast cancer one. The same tendency was described in Ref. [43], where the authors evaluated the internalization of micron-sized (PAH/PSS)_4_ capsules by different normal and cancer cell lines after 24 h of co-incubation with the capsules. Additionally, we found that Raw 264.7 cells more actively captured 2.7 μm capsules at all time points (Figure 4a), whereas 4T1 cells demonstrated the same tendency for just the first six hours, and then the number of cells that internalized 1 μm carriers was higher (Figure 4b).

Since macrophages are characterized by higher phagocytic activity in comparison with tumor cells. The Raw 264.7 cell line was chosen as a model for the verification of the capability of the magnetic trapping of cells from the flow. Additionally, we chose 2.7 μm capsules loaded by MNPs in three freezing/thawing cycles for these experiments since their internalization efficiency as well as magnetic moment are higher than for 1 μm capsules. The flow velocity was around 5 mm/s to reduce viscous drag force since the size (around 13 μm) and weight of Raw 264.7 cells are much higher than the largest investigated capsules (5.5 μm).

The magnetic capturing of cells incubated with five capsules per cell showed a four times increase in the number of cells adsorbed on the internal side of the flow cell’s wall in comparison with the control (electrostatic adhesion of cells) (Figure 5a). The increase in microcapsules’ concentration to up to 10 capsules per cell led to a seven-fold rise in the captured cell number compared with the control (Figure 5a).

The evaluation of cell membrane integrity after magnetic capture showed an increase in the number of damaged cells depending on the capsule per cell concentration. Thus, the magnetic capture of cells that underwent 24 h of co-incubation with capsules with a concentration of five capsules per cell led to a decrease in the number of viable cells by 13% in comparison with the control, and co-incubation with 10 capsules per cell led to a 15% decrease (Figure 6b).

## 3. Discussion

A magnetic field seems to be the most promising tool for noninvasive remote control over magnetically sensitive objects within the body due to its high penetration depth into biological tissues [28]. With this in mind, the capability to precisely manipulate micron-sized objects possessing magnetic properties, as demonstrated in this study, is also a great advantage [26,31,44,45]. Magnetic capsules tend to move according to the magnetic field gradient in the direction of its enhancement which facilitates capsule accumulation in the desired area with high precision [46,47,48]. Moreover, the application of an alternate magnetic field is able to induce the release of the cargo from the capsule by means of the mechanical or thermal effects of an oscillating field [49,50,51,52]. All of the above factors make the magnetic field an attractive candidate for solving issues of remotely controlled drug delivery and selecting prognostic markers from the bloodstream.

The magnetic trapping of target objects from the flow and their holding in the region of interest is not a trivial issue and requires detailed investigation. The obtained data demonstrated the considerable influence of the object properties, such as size and weight, on its capture efficiency. In conjunction with the previously obtained data about the circulation duration of micron-sized capsules in the vascular system of mice and their cyto- and hemotoxicity, the profile of an effective system was formed.

According to Ref. [11], 1 μm and 2.7 μm magnetic capsules synthesized from biocompatible polyelectrolytes such as Parg and DS and loaded with MNPs in three freezing/thawing cycles do not affect macrophage viability and RBCs’ membrane integrity at a concentration of up to 50 particles per cell. A small decrease for both parameters was found for the application of 5.5 μm capsules at concentrations of 10 capsules per cell and higher, which points to the advantage of the capsules of 2.7 μm and a smaller average diameter. Additionally, the circulation duration of capsules shortens with the enlargement of their size, which also reflects the advantage of small capsules. However, the first data about the efficiency of the magnetic trapping of capsules from the flow at a 30 μL/min flow rate showed a tendency of more efficient capture with the increase in capsule size and the number of freezing/thawing cycles [11]. It correlates with the MFM data presented in the current study, which show the same tendency of a capsule magnetic moment increase (Figure 1). It should be related to the payload capacity rise of a single vaterite particle used as a template for capsule synthesis following the enlargement of its surface area and internal volume driven by its increased size [32,33]. However, close consideration of capsules’ capture process at different flow velocities brought to light the importance of both their size and weight. We found an atypical decrease in the capture efficiency of 5.5 μm capsules loaded through three freezing/thawing cycles when compared with 2.7 μm ones despite the higher magnetic moment.

The forces affecting magnetic capsules passing through the flow cell with the adjacent permanent magnet include the viscous drag force and magnetic force [31]. The mathematical model (6) describing both of them showed that the capsule size, as well as weight, significantly influence the strength of drag force. An increase in a capsule’s size and weight leads to the enhancement of that force. Since successful magnetic trapping of the capsule in the flow is possible in case of equity or the prevalence of the magnetic force over drag, the capsule’s size and weight thresholds should exist for a certain flow velocity, behind which the capsules’ capture is impossible. It could also be a reason for the atypical tendency observed.

The influence of the “protein corona” formation on the magnetic trapping of capsules in the flow is another important aspect of this issue since protein adsorption on the capsule surface can affect its charge, size, weight, and tendency to aggregation. According to Ref. [53], the “protein corona” formation process mainly depends on the surface charge of objects. The last layers of Parg/DS or PAH/PSS capsules considered in the current study are formed by negatively charged DS and PSS, respectively [54,55]. Previously, Zhou J. Deng et al. demonstrated that different isoforms of Fibrinogen, Albumin, and Apolipoprotein A1 are the main plasma protein components that adsorb on the nanoparticle surface independently of their surface charge [56]. Therefore, we suggest that the “protein corona” formed on the surface of capsules under consideration would be caused by the same proteins. According to Ref. [57], Fibrinogen is a negatively charged protein highly contributing to nanoparticle aggregation in plasma. Our previous investigation demonstrated an increase in the number of Parg/DS microcapsule aggregates over time in plasma [27], which corresponds with the data presented for nano-sized objects. However, we also found that the numerous blood cells prevent capsule aggregate formation due to the isolation of capsules from each other. Therefore, we can assume enhanced capsule aggregate formation during magnetic trapping from the blood caused by the “protein corona” effect principally at the site of magnetic field application (area of magnetic microcapsules’ accumulation) since the probability of the aggregation of capsules smoothly distributed in blood would be extremely low. The enlargement of a capsule’s weight and size caused by the adsorption of “protein corona” on its surface can lead to a slight decrease in the capsule’s magnetic trapping efficiency due to the enhancement of the strength of the drag force affecting it. However, the adsorption of such an extra layer on the capsule’s surface cannot significantly change its hydrodynamic properties to make the capsules’ magnetic trapping in flow impossible.

Additionally, it is worth noting that the adsorption and accumulation of capsules on a blood vessel’s or a flow cell’s wall leads to a decrease in the vessel lumen, thereby resulting in subsequent local flow velocity acceleration, which in turn enhances the strength of the drag force (1). It can also cause the results described above since the accumulation of 5.5 μm capsules on the internal wall of the flow cell decreases its lumen more significantly compared with the accumulation of 2.7 μm capsules. Thus, we can assume the existence of a normal distribution for capsules’ capture efficiency depending on their size, the extremum for which we observed with 2.7 μm capsules loaded by MNPs in three freezing/thawing cycles. For capsules loaded with a lower number of MNPs, we are unlikely to be able to observe the same normal distribution since the capsules’ magnetic moment is not strong enough to accumulate the substantial aggregate for the effect of flow acceleration to begin to make a difference.

Additionally, this finding elucidates the necessity for the following study of capsule aggregates behavior in the vasculature system, since their formation at the site of magnetic force application presents a potential threat of emboly. Previous studies have demonstrated the high potential of easily deformable hollow drug delivery systems for systemic administration due to their mechanical properties and insignificant reversible effects on the blood flow velocity in vital organs [17,58]. However, the evaluation of the flow velocity in mouse kidneys and liver after the tail vein injection of 5 and 50 million capsules resuspended in 200 μL of saline showed more pronounced changes in the flow rate for a higher concentration. This can be explained by the higher number of capsule aggregates formed during administration. Theoretically, the best option for microcapsule systemic administration is the slow injection of a relatively small number of capsules resuspended in the maximal available volume of saline, since it decreases the probability of capsule aggregate formation. At the moment of a capsule’s entrance into the blood flow, it is additionally diluted by blood and single capsules are isolated from each other by numerous blood cells [27], which enhances the capsules’ aggregate formation and decreases the risk of emboly. Therefore, previously observed flow rate changes in organs [17] indicate a partial vascular occlusion. A large aggregate of capsules formed during their targeting of the affected area by the external source of the magnetic field that can be subsequently swept up by the blood flow may lead to much more serious sequelae including myocardial infarction [59] and acute stroke [60]. We can expect the desegregation of such large emboli under the blood pressure to small ones or even single capsules; however, the contribution of capsule magnetization intensity and the “protein corona effect” [27] to the maintenance of aggregate integrity has not been clearly investigated.

The magnetic trapping of prognostic markers such as CTCs is a much more difficult issue than the capture of a single capsule since we must consider the interaction of the cell with magnetic labels. A cell, in comparison with a capsule, does not possess its own magnetic moment. It is determined by the sum of magnetic moments of objects that interact with the cell, which additionally sophisticates the investigation and modeling of this process.

Theoretically, the interaction of the cell with a smaller object could result in its adsorption on the cell’s surface or its uptake [11,61]. All currently existing kits for the magnetic separation of different cell fractions from the blood sample are based on the IHC interaction of antibodies linked with a magnetic object and antigen on the cell membrane [38,39,40], since this kind of interaction is highly selective. However, such a model of cell-label interaction further improves the investigation of the magnetic trapping of labeled cells from the flow since we must consider the sail-effect and the random possibility of IHC bond breaking. In this view, cells that internalize magnetic microcapsules are a much more convenient model.

The comparative study of the capsules’ capture efficiency by immune and cancer cells showed the predominance of macrophages despite cancer cells being larger in size. This effect was caused by the macrophages’ activation as a response to interaction with magnetic polyelectrolyte microcapsules and the initiation of their phagocytosis [62,63].

The evaluation of the magnetic trapping efficiency of Raw 264.7 cells co-incubated with different numbers of magnetic microcapsules showed the tendency to increase in the number of captured cells following an increase in capsule concentration. This is caused by the enlargement of a number of cells that internalize a sufficient number of capsules to be captured by a magnetic field. Since the macrophage size (around 13 μm) is 4.8 times higher than the 2.7 μm capsule, the viscous drag force affecting the cell is also equivalently higher at a certain flow velocity (1). Thus, a minimal cut-off number of capsules should exist, the internalization of which is required for the magnetic trapping of a single cell since the cell’s magnetic moment can be considered as a sum of the magnetic moments of the internalized capsules.

The examination of membrane integrity after the magnetic separation of cells showed an increase in the number of damaged cells following an increase in capsules’ concentration. This increase is probably caused by the damage of the cell membrane during the magnetic trapping of the cell by internalized magnetic capsules. The external source of the magnetic field is capable of catching and strongly holding capsules in the area of the strongest magnetic field [11,26]. Therefore, a cell moving under the influence of the viscous drag force of the flow is stopped due to internalized capsules which play the role of anchors. At the same time, the energy of the drag force converts into the pressure applied on the cell membrane by a magnetically captured capsule. Considering the viscous drag force as a flow pressure exerted on the cell surface area allows us to suppose that the capsule pressure exerted on the internal side of the cell membrane in the opposite direction to the flow will be as many times as high as the capsule area that is in contact with the membrane and less than the area of the cell affected by the flow pressure. Therefore, the application of a magnetic field that is strong enough to capture the capsule inside the cell and hold it further can lead to cell membrane damage by the capsule under the flow drag force influence and consequent cell death. Dealing with the problems of targeted delivery of such magnetic labels inside the pathological cells circulating in the vasculature system will further prospects for their selective damaging and killing, which is a challenging issue in modern medicine.

## 4. Materials and Methods

### 4.1. Materials

Iron (III) chloride hexahydrate (99.8%), iron (II) chloride tetrahydrate (99.8%), sodium hydroxide (99.8%), citric acid (99.8%), sodium carbonate, calcium chloride, sodium chloride, poly-l-arginine hydrochloride (Parg, Mw = 15–70 kDa), dextran sulfate sodium salt (DS, Mw = 100 kDa), bovine serum albumin (BSA, lyophilized powder), poly(allylamine hydrochloride) (PAH, Mw = 50 kDa), poly(sodium 4-styrenesulfonate) (PSS, Mw = 70 kDa), ethylenediamine tetraacetic acid disodium salt (EDTA), phosphate-buffered saline (PBS, 0.01 M), ammonium rhodanide (NH4SCN), and Dulbecco’s modified Eagle’s medium (DMEM) with a high glucose content were obtained from Sigma-Aldrich (St. Louis, MO, USA). Glycerin was purchased from Reachem (Moscow, Russia). Fetal bovine serum (FBS), penicillin–streptomycin solution, 0.25% trypsin solution with 0.02% EDTA, DAPI Solution (1 mg/mL), Alexa Fluor 488 Phalloidin, and the 0.4% Trypan blue solution was obtained from Thermo Fisher Scientific (Waltham, MA, USA). All chemicals were used without additional purification. Deionized water (specific resistivity higher than 18.2 M∙Ω∙cm) from the Milli-Q Direct 8 (Millipore, Merck KGaA, Darmstadt, Germany) water purification system was used to prepare all solutions.

### 4.2. SPIM-Fluid Imaging Flow Cytometer

In this work, we use a modified version (Figure 2a) of the SPIM-Fluid imaging flow cytometer [64], where the fluorescence of FITC and TRITC dyes is excited by the radiation of one 488 nm laser. A scheme of the cytometer consists of several logical blocks, including the following:A light-sheet illumination subsystem: CW diode lasers of 488 nm (Cobolt MLD 06-01, Hübner Photonics, Kassel, Germany), 3.5× beam expander (formed by the pair of lenses: LA1131-A and LA1229-A, Thorlabs, Newton, NJ, USA), the light-sheet forming system that consists of a cylindrical lens (f = 50 mm, LJ1695RM-A, Thorlabs, Newton, NJ, USA) and an objective (4×, NA = 0.13, CFI Plan Fluor, Nikon, Tokyo, Japan).A modified image detection subsystem: spatial splitter of FITC and TRITC channel images on the camera (Dhyana 400BSI, Tucsen) placed between an objective (MO, 10×, NA = 0.3, CFI Plan Fluor, Nikon, Tokyo, Japan) and a tube lens (LBF254-200-A, Thorlabs). The spatial splitter consists of a dichroic mirror (T560lpxr, Chroma), 2 broadband silver mirrors, fluorescence filters (FITC Emission Filter-MF530-43, TRITC/CY3.5 Emission Filter-MF620-52, Thorlabs, Newton, NJ, USA), and a tilted beam splitter (BSW10R, Thorlabs, Newton, NJ, USA).A flow cell: UV Quartz clear flow-through cuvette (526UV0.25, FireflySci, New York, NY, USA) hermetically connected with a syringe pump (AL-1000, World Precision Instruments, Sarasota, FL, USA) by plastic tubing. The source of the magnetic field is the permanent rare earth magnet with a custom magnetic field concentrator that provides a magnetic field strength of up to 0.3 T in the position at the edge of the flow cell channel.

### 4.3. Preparation of Magnetic Fluorescent Polyelectrolyte Microcapsules

#### 4.3.1. Syntheses of Calcium Carbonate Microparticles of Various Sizes

Calcium carbonate (CaCO_3_) microparticles with different diameters were synthesized by mixing equimolar calcium chloride (CaCl_2_) and sodium carbonate (Na_2_CO_3_) solutions. Minor corrections were made in the standard protocol of CaCO_3_ particles synthesis [65] to obtain particles with average sizes of 1, 2.7 and 5.5 μm, as was described previously in Reference [11]. Briefly, 2.7 and 5.5 μm microparticles were obtained in the water subphase at a slow agitation rate (500 rpm for 2.7 μm particles, and 100 rpm for 5.5 μm particles) by mixing 1 M salt solutions, whereas 1 μm microparticles were synthesized in ethylene glycol using 0.1 M solutions of the salts. The obtained microparticles were collected by centrifugation and thoroughly washed with deionized water.

#### 4.3.2. Magnetite Nanoparticles Synthesis

The method of MNPs’ synthesis was previously described in detail [66]. Briefly, MNPs were synthesized by the chemical precipitation of iron (II) and iron (III) salts in the alkaline medium in the inert atmosphere of nitrogen at 40 °C followed by stabilization with 0.019 M citric acid. Then, the stabilized MNP suspension was dialyzed for 3 days in deionized water at a slow agitation rate. The concentration of MNPs was evaluated using the colorimetric titration method and found to be 1 mg/mL.

#### 4.3.3. Preparation of the BSA-FITC Conjugate

BSA-FITC conjugate was prepared according to the protocol previously described in Ref. [14]. Briefly, 1 mL of FITS solution (1 mg/mL in DMSO) was mixed with 40 mL of BSA solution (4 mg/mL in 0.1 M PBS buffer, pH 8), and incubated with gentle stirring at 4 °C for 12 h. Then, the prepared BSA-FITC conjugate solution was dialyzed for 3 days against deionized water to remove the unreacted dye.

#### 4.3.4. Loading of Calcium Carbonate Microparticles with MNPs

The loading of MNPs into microcapsules was achieved by the resuspension of 40 mg CaCO_3_ microparticles in MNPs colloid suspension (1 mL, 1 mg/mL) and freezing at a continuous agitation rate, as was described in Ref. [19]. The amount of freezing/thawing cycles varied from 1 to 3 to obtain microparticles containing different numbers of MNPs. Synthesized CaCO_3_ microparticles loaded with MNPs were employed as templates for the further assembly of polyelectrolyte microcapsules via the layer-by-layer (LbL) technique.

#### 4.3.5. Formation of Multilayer Shells

The capsules were prepared via the layer-by-layer technique using the obtained calcium carbonate particles as a sacrificial template. Multilayer shells comprising 4 bi-layers of polyelectrolytes represented either by PSS/PAH or Parg/DS were assembled on the surface of CaCO_3_ particles loaded with MNPs. To visualize the capsules, a layer of BSA-FITC was added into the shells. Polyelectrolytes were alternatively adsorbed from 2 mg/mL (for 1 μm capsules due to increased surface area) and 1 mg/mL (for 2.7 and 5.5 μm capsules) aqueous solutions, which also contained 0.5 M NaCl, onto 40 mg of CaCO_3_ per sample. The following compositions were designed: (PSS/PAH)(PSS/BSA-FITC)/(PSS/PAH)_2_PSS and (Parg/DS)(BSA-FITC/Parg)/(Parg/DS)_2_. The obtained coated particles were treated with 5 mL of 0.2 M EDTA to remove CaCO_3_ cores, resulting in hollow polymeric capsules.

### 4.4. Characterization of Magnetic Microcapsules

#### 4.4.1. Magnetic Force Microscopy Characterization of Magnetic Microcapsules

MFM characterization of 1, 2.7 and 5.5 μm PAH/PSS magnetic microcapsules containing different concentrations of MNPs was performed using NTEGRA Spectra AFM (NTMDT, Zelenograd, Russia) equipped with magnetically coated cantilevers NSG01/Co (NTMDT, Zelenograd, Russia) with the following nominal parameters: length L = 130 ± 5 μm, width w = 35 ± 3 μm, thickness t = 2 μm, measured spring constant k_c_ = 4.9 N/m, first free resonance in air f_0_ = 142 kHz. The two-pass MFM technique with a 15 nm lift height for the second pass was used. All MFM images were analyzed using Gwyddion software [67]. The magnetic force acting on the cantilever was measured as a phase signal.

#### 4.4.2. The Efficiency of the Magnetic Trapping of Capsules

The magnetic trapping efficiency of 1, 2.7 and 5.5 μm PAH/PSS capsules containing different concentrations of MNPs was evaluated depending on the flow rate using a SPIM-Fluid-based custom imaging flow cytometer (Figure 2c). To perform this, the suspension of polyelectrolyte microcapsules was passed through the flow cell with the adjacent permanent magnet. Laminar suspension flow was provided by a syringe pump (AL-1000, World Precision Instruments, Sarasota, FL, USA). Objects captured by the magnetic field were counted using the computer vision method described below in the data analysis section.

### 4.5. Cell Model

#### 4.5.1. Cells Preparation

The murine macrophages (Raw 264.7) and breast cancer (4T1) cell lines were provided by the Science Medical Center, Saratov State University, Russia. Both cell cultures were grown in Dulbecco’s modified Eagle’s medium containing 4500 mg/L of glucose supplemented with 10% of FBS, and 1% of penicillin-streptomycin at a humidified atmosphere of 5% CO_2_ at 37 °C. The growth medium was replaced every 3 days. Cells that reached 90% confluence were detached using 0.05% trypsin with EDTA and counted using the Countess automated cell counter (Thermo Fisher Scientific, Waltham, MA, USA).

#### 4.5.2. Comparative Evaluation of Capsules’ Internalization Efficiency

The efficiency of 1 and 2.7 μm Parg/DS microcapsule internalization after 3, 6, 12 and 24 h of co-incubation was investigated for Raw 264.7 and 4T1 cell lines. To perform this, magnetic polyelectrolyte microcapsules were labeled with FITC and added to cells at a concentration of 10 capsules per cell. The visualization of cells was performed using the ImageStream X Mark II Imaging Flow Cytometer (Luminex, Austin, TX, USA), as was described in Reference [11]. Briefly, cells were washed from uncaptured capsules with PBS, fixed with 10% neutral buffered formalin, and stained with 1 μg/mL of Alexa Fluor 488 phalloidin and 1 μg/mL of DAPI. Then, cells were resuspended in PBS and measured by performing imaging flow cytometry (Figure 4c). For the analysis of the obtained data, the “Internalization” and “Spot counter” tools of IDEAS 6.2 software (Luminex) were used.

#### 4.5.3. The Efficiency of the Magnetic Capture of Cells

The magnetic capture efficiency of Raw 264.7 cells incubated without/with Parg/DS magnetic microcapsules added in concentrations of 5 and 10 capsules per cell for 24 h was evaluated using a SPIM-Fluid-based custom imaging flow cytometer and the ImageStream X Mark II Imaging Flow Cytometer (Figure 2d). For this, cells suspended in PBS were passed through the flow cell with the adjacent permanent magnet. Then, magnetically captured cells were collected, stained with 1 μg/mL of Hoechst 34580 for 30 min, and enumerated using an Imaging Flow Cytometer (Figure 5b). All values were normalized for the control sample value (cells incubated without capsules), which showed the magnet independent adhesion of cells on the flow cell wall.

#### 4.5.4. Evaluation of the Membrane Integrity of Magnetically Captured Cells

The cell membrane integrity of magnetically captured Raw 264.7 cells was estimated by performing the double-staining method using the combination of membrane-permeable Hoechst 34580 and impermeable Propidium Iodide (PI) dyes. For this, magnetically captured cells were stained with 1 μg/mL solution of both dyes in PBS for 30 min, then triply washed with PBS, and analyzed using an ImageStream X Mark II Imaging Flow Cytometer (Figure 6a). Intact Raw 264.7 cells were used as a positive control, and cells fixed by adding 30% *v*/*v* of ethanol were used as a negative one. Additionally, we used reference samples that were incubated with different concentrations of capsules but did not undergo magnetic separation.

### 4.6. Data Analysis

#### Magnetic Capsules’ Capture

The method for detecting fluorescent objects in the detection stream is based on the selection of the object’s contour against the background. When analyzing the video stream, four cases were observed. The first is the usual movement of clearly distinguishable objects. The second is the formation of large aggregates with intense fluorescence. The third one is the case of particles that are weakly fluorescent or located near the wall of the cuvette opposite the detector. The fourth stage is an intermediate stage in the formation of aggregates when the particles are not yet sticking together, but light traces are already superimposed on each other. For stronger edge detection, we applied the Sobel operator to the images (which makes the particles more distinguishable in the third case). For the first case, we can find all the contours in a single frame and calculate their areas using the implementations of the Suzuki algorithm and Green’s formula in the OpenCV library. Based on the values of these areas, we estimate the average contour area corresponding to a separate object. We then subtract adjacent frames to reveal moving objects and count them by the number of edges (one edge per object). For the case of sticking objects, we determine the area of the bright spot and divide it by the calculated average area of the contour of one object. Since the areas of the spots change in proportion to the number of adhered (or washed away) objects, approximate calculations of the number of objects in the frame can be obtained. For a more accurate analysis of separate particles moving and sticking to the magnet in the fourth case, we additionally find the centroids of the particle contours and monitor the change in their coordinates on neighboring frames. Despite their simplicity, these algorithms show good object counting accuracy. This was confirmed by a video obtained from a sample with a known number of objects. The algorithms found 89–93% of objects, depending on the presence of a cluster of objects.

## 5. Conclusions

The magnetic trapping of objects in a fluid flow underlies biomedical approaches such as magnetically controlled targeted drug delivery through systemic administration and the isolation of magnetically labeled prognostic markers from the bloodstream. The introduction of such approaches into medical practice requires a thorough study of the magnetic trapping process. Here, we evaluated capsules’ capture efficiency depending on their size, magnetic moment, and velocity of a suspension flow. We tested the ability of magnetically separating the cells from the flow and then examined the integrity of their cell membranes. The evaluation of capsules’ capture efficiency showed a tendency of growth with an increase in the capsule size and the number of freezing/thawing cycles, which vaterite templates underwent during capsule formation. The viscous drag force acting upon the capsules in the flow has also been shown to highly depend on their size and weight, which imposes some restrictions on the developed drug delivery system. The magnetic trapping of Raw 264.7 cells co-incubated with different concentrations of magnetic capsules showed a tendency of an increase in the number of captured cells as of the capsules’ concentration increased. A decrease in cell membrane integrity was also observed for magnetically trapped cells which opens the prospect for the noninvasive destruction of pathological cells circulating in the vascular system. Based on the obtained data, the main features that assist in the magnetic capture of objects from the flow were identified, which can help to form a detailed profile of an effective system for magnetically controlled drug delivery and the extraction of prognostic markers from the blood flow.

## Figures and Tables

**Figure 1 molecules-27-06073-f001:**
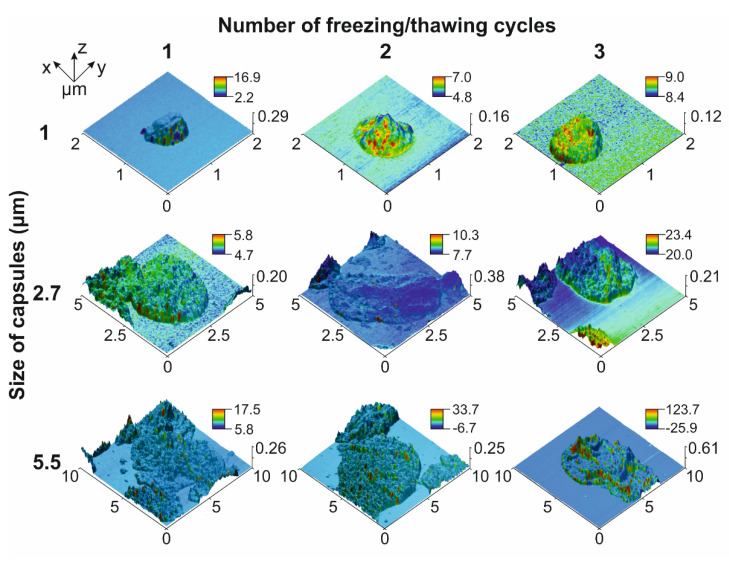
MFM images of 1, 2.7 and 5.5 μm PAH/PSS capsules loaded with different amount of MNPs. MFM phase signal is presented in the color bar (deg).

**Figure 2 molecules-27-06073-f002:**
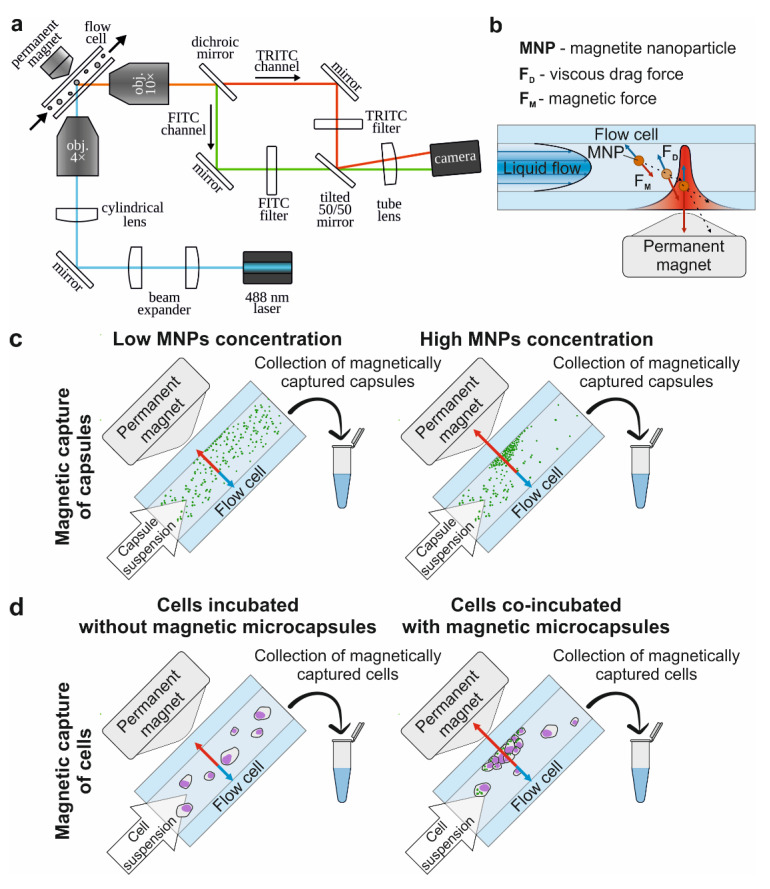
The principal scheme of the study. (**a**) Experimental setup of the SPIM-Fluid imaging flow cytometer equipped with a permanent magnet, providing the visualization of fluorescently labeled objects passing through the flow cell with a 488 nm laser and camera and the magnetic trapping of objects in flow using 0.3 T magnet equipped with conical concentrator. (**b**) Scheme of magnetic capturing of the object in the flow. (**c**) Scheme of the evaluation of the magnetic capsules’ capture efficiency. (**d**) Scheme of the evaluation of the magnetic capture efficiency of cells. Blue arrows indicate the vectors of viscous drag force action, red arrows indicate the vectors of magnetic force action.

**Figure 3 molecules-27-06073-f003:**
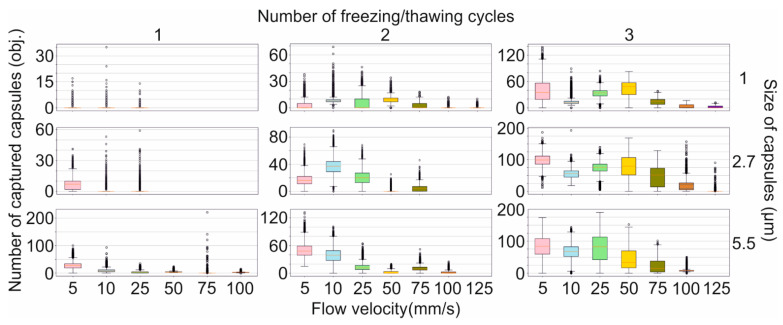
The efficiency of magnetic capture of differently sized polyelectrolyte microcapsules from the flow depending on the amount of loaded magnetite and the flow velocity.

**Figure 4 molecules-27-06073-f004:**
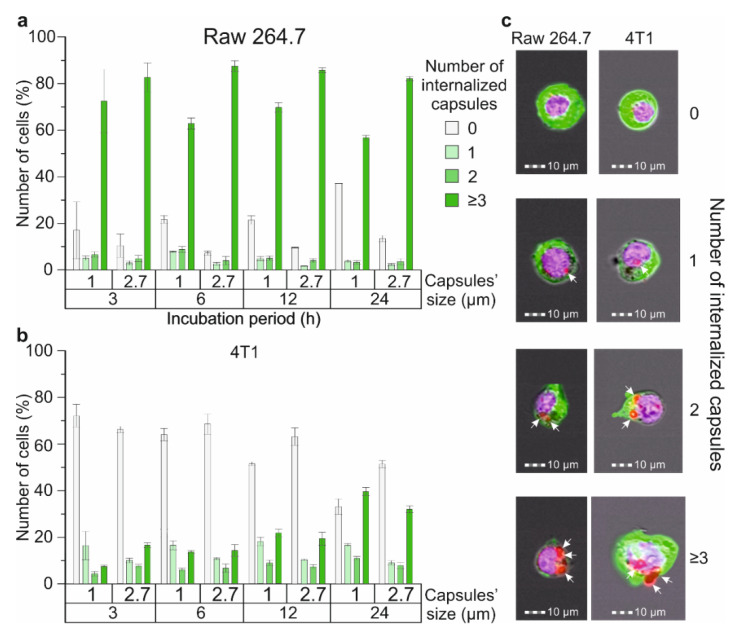
The efficiency of 1 and 2.7 μm magnetic polyelectrolyte microcapsule internalization by Raw 264.7 and 4T1 cell lines: the number of (**a**) Raw 264.7 and (**b**) 4T1 cells that captured one or several capsules with an average diameter of 1 and 2.7 μm, depending on time; (**c**) Brightfield/fluorescent images of Raw 264.7 and 4T1 cells that captured 1, 2, 3 and more microcapsules (white arrows indicate internalized capsules).

**Figure 5 molecules-27-06073-f005:**
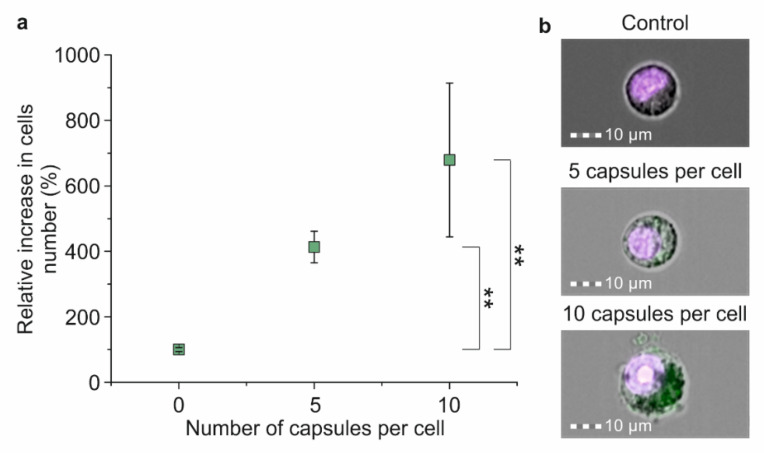
The efficiency of the magnetic separation of Raw 264.7 cells co-incubated with magnetic polyelectrolyte microcapsules: (**a**) Relative increase in captured cell number depending on the amount of co-incubated microcapsules; (**b**) Joint brightfield/fluorescent images of Hoechst 34580 stained cells co-incubated with FITC-labeled microcapsules in concentrations of 0, 5 and 10 capsules per cell for 24 h. Double (**) asterisks indicate the significance (*p* < 0.01).

**Figure 6 molecules-27-06073-f006:**
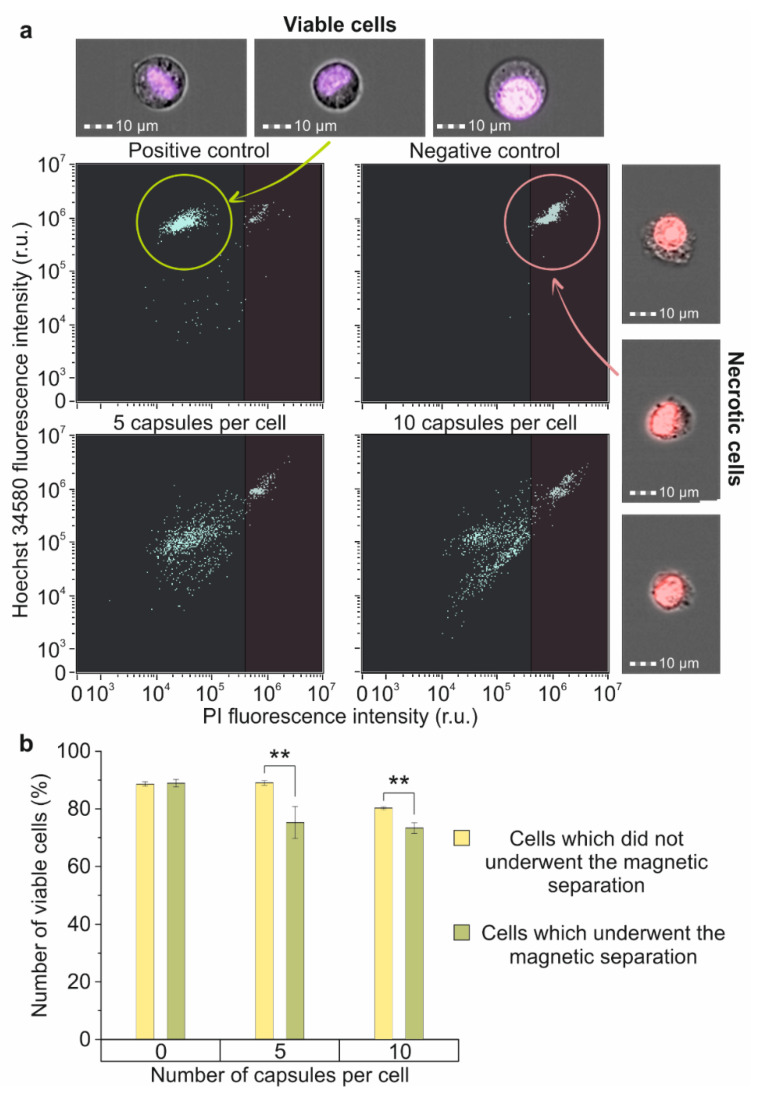
The impact of the magnetic separation on membrane’s integrity and viability of Raw 264.7 cells: (**a**) Dot plots and joint brightfield/fluorescent images of Hoechst 34580/PI stained cells: intact cells (positive control-left upper part), fixed cells (negative control-right upper part), and magnetically separated cells, which have been incubated with 5 (left lower part) and 10 (right lower part) microcapsules per cell for 24 h; (**b**) Percent of viable cells in samples incubated with 0, 5 and 10 microcapsules per cell that underwent/did not undergo magnetic separation. Double (**) asterisks indicate the significance (*p* < 0.01).

## Data Availability

Not applicable.

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
