# Peer review of "The Influence of Magnetic Composite Capsule Structure and Size on Their Trapping Efficiency in the Flow"

_molecules, 2022, doi:10.3390/molecules27186073_

Round 1
Reviewer 1 Report
The paper is devoted to the study of an interesting aspect of polyelectrolyte microcapsules application, in particular the influence of magnetic composite capsule structure and size on their trapping efficiency in the flow. In general, the paper concerns a topical issue of medicine. However, the paper has major problems which should be discussed:
The size of figures should be enlarged. Also the quality of figures should be improved too.
The figure 5. The error bars of 3rd dot (10 capsules per cell) are too large. With that connection question is how the the significance were calculated of results.
Does increasing the number of freeze and thaw cycles increase the internal volume of the capsule, or does just increasing the size of the capsule increase the internal volume? Please clarify this point in the results (lines 115-125).
What could be the reason of changes of internalization tendency of microcapsules tendency for 4T1 cells?(lines 240-243)
How will the presence of proteins in the bloodstream affect the captchering of capsules? As is known, the surface of capsules is capable of sorbing protein from solution, which affects the charge of the capsule surface, their aggregation, adhesion, and hydroaerodynamics.Discuss it in connection of your provided results, please.
It is poorly discussed the behaviour of microcapsules at the bloodstream. (macrophage internalization, adhesion of protein red cells, white cells on the microcapsules surface, blockage of blood vessels by polyelectrolyte microcapsules, the dependence of microcapsule's behaviour from thier size).
Figure 4. Why the microcapsules with 5.5 size have not studied?
Lines 105-107. It is will be usefull to detail about safety limitations.
Line 115. What the composition of microcapsules (pss/pah)? Number of layers.
Line 115. What the error bars of polyelectrolyte microcapsules sizes?
Line 120. Present main result or the data presented in [13,29].
lines 153-158. I will strongly recomend to move that to Materials and methods.
The bibliography data used in the section results and discussion i recomend to move to introductionsection.Because the large amount of text makes it difficult to read and understand the results.
Reviewer 2 Report
An interesting article devoted to assessing the influence of the size and magnetic properties of magnetic polyelectrolyte microcapsules on the possibility of their concentration in a constant magnetic field. As the authors have shown, these capsules can be used for magnetic separation of cells. The work is new and may be of interest to specialists working in the field of targeted drug delivery systems and cell technologies. In general, I see no obstacles to the publication of the work, however, the authors need to correct a number of shortcomings:
1. The meaning of the overline notation in equations is not clear. If it is used to denote vectors, it should be present or absent over both and in equation (1) and all “nab” notations should be overlined.
2. It is not clear what the relation authors assumed between equations (3) and (4). One can guess, based on equation (4), that the “+” sign was supposed to be used in equation (3) instead of the dot.
3. The dashes in lines 138, 142, 144, and 145 should be replaced by the word “is”.
4. The authors need to justify the possibility of using magnetic force microscopy to assess those magnetic properties of capsules that are taken into account in formulas 3 and 4. (?0 - initial magnetization, ????- magnetic susceptibility of the capsule).
5. It is also advisable to give (possibly in the appendix) the numerical values of the magnetic properties of the capsules, and not only for the PAH/PSS capsules, but also for the Pars/DS capsules
6. It is necessary to describe the principle of working of the experimental setup shown in fig. 2a. (better - in the caption to figure 2).
7. In the text (in the caption to Fig. 2 or in the “Materials and Methods” section), describe the content of Fig. 2b.
8. In fig. 2c and 2d clarify the direction of the Stokes force. With a uniform motion of a particle in a fluid flow, this force is equal to zero. With an axial moving under the action of a magnetic force, it is directed opposite to this force.
Round 2
Reviewer 1 Report
Authors significantly improved the article and after minor correction it may be accepted:
1. Authors described the disadvantages of particles application inside a bloodstream, but PMC have thesame problems like other partiles. They didn't described advantages of polyelectrolyte microcapsules. If authors discribed the disadvanteges of particles then it is nessecary to discribed prons and cons of polyelectrolyte microcapsules.
2. Authors have not investigated the internalization efficiency of 5.5 µm capsules by murine breast cancer cells because a problem with the analysis of data for 5.5 µm capsules. Why authors researched other parameters for 5.5 µm capsules if they have bad data in internalization?
